# Infection and telomere length: a systematic review protocol

Louis Tunnicliffe  ,[1] Rutendo Muzambi  ,[1,2] Jonathan W Bartlett,[1] Laura Howe,[3] Khalid Abdul Basit,[1] Charlotte Warren-Gash  [1]

[1]Faculty of Epidemiology & Population Health, London School of Hygiene & Tropical Medicine, London, UK
[2]Faculty of Epidemiology and Biostatistics, Imperial College London, London, UK
[3]MRC Integrative Epidemiology Unit, Department of Population Health Sciences, Bristol Medical School, University of Bristol, Bristol, UK

**Correspondence to**
Mr Louis Tunnicliffe;
louis.tunnicliffe@lshtm.ac.uk

## ABSTRACT

**Introduction** Telomeres are a measure of cellular ageing with potential links to diseases such as cardiovascular diseases and cancer. Studies have shown that some infections may be associated with telomere shortening, but whether an association exists across all types and severities of infections and in which populations is unclear. Therefore we aim to collate available evidence to enable comparison and to inform future research in this field.

**Methods and analysis** We will search for studies involving telomere length and infection in various databases including MEDLINE (Ovid interface), EMBASE (Ovid interface), Web of Science, Scopus, Global Health and the Cochrane Library. For grey literature, the British Library of electronic theses databases (ETHOS) will be explored. We will not limit by study type, geographical location, infection type or method of outcome measurement. Two researchers will independently carry out study selection, data extraction and risk of bias assessment using the ROB2 and ROBINS-E tools. The overall quality of the studies will be determined using the Grading of Recommendations Assessment, Development and Evaluation criteria. We will also evaluate study heterogeneity with respect to study design, exposure and outcome measurement and if there is sufficient homogeneity, a meta-analysis will be conducted. Otherwise, we will provide a narrative synthesis with results grouped by exposure category and study design.

**Ethics and dissemination** The present study does not require ethical approval. Results will be disseminated via publishing in a peer-reviewed journal and conference presentations.

**PROSPERO registration number** CRD42023444854.

## INTRODUCTION
### Rationale

Telomeres are structures found at the ends of chromosomes which are composed of repetitive DNA sequences and protective proteins. Their primary role is to shield the genomic DNA from being recognised as damaged or broken to prevent processes such as DNA end-joining, DNA recombination, or DNA repair that could lead to chromosome instability.[1]

The DNA replication machinery in cells cannot fully copy the DNA at the extreme ends of linear chromosomes, which results in the gradual shortening of chromosome ends with each cell division.[1] Eukaryotic

cells address this via an enzyme called telomerase which acts to replenish the chromosome ends.[2] However, in many human cell types, the levels of telomerase (or its activity on telomeres) are limited. This combined with factors such as nuclease action, chemical damage and DNA replication stress results in the continuous shortening of telomeres throughout a person's lifespan. For this reason, telomere length is used as a measure of biological ageing.[1]

### STRENGTHS AND LIMITATIONS OF THIS STUDY

⇒ This study will be conducted in adherence to the established Preferred Reporting Items for Systematic Reviews and Meta-analyses Protocols statement.
⇒ Study selection, data extraction, risk of bias assessment and certainty assessment will be performed by two independent reviewers using well-established guidelines and methods.
⇒ We will search a range of relevant databases including published and grey literature.
⇒ One limitation is that suspected heterogeneity with respect to exposure type and outcome measurement may mean a meta-analysis is not possible.

When telomeres reach a critical length or experience significant damage, a prolonged DNA damage response is triggered. This results in changes to gene expression patterns and leads to cellular senescence.[3] The specific outcomes of senescence are thought to vary depending on cell type.[1] It has been extensively documented that inflammation plays a significant role in the progression of diseases like cardiovascular disease, chronic kidney disease and Alzheimer's disease.[4] Given that immune cell senescence induces pro-inflammatory processes, telomere attrition in immune cells becomes relevant to the development of these conditions.[1] Furthermore, there is evidence to suggest that telomere shortening is associated with increased incidence of various diseases including Alzheimer's disease[5] and cardiovascular disease[6] even after adjusting for age. The idea that shorter telomeres are a potential risk factor

for age-associated diseases is reinforced by the fact that inherited telomere syndromes, where individuals are genetically pre-disposed to have short telomeres, are characterised by phenotypes of accelerated ageing, including a host of age-associated diseases.[7]

Telomere length has been shown to be associated with lifestyle and environmental factors.[1] For example, early-life connections between stress and telomeres are evident.[8] Infections are another factor which could influence telomere length via pathways such as inflammation and oxidative stress.[9 10] However, there is a lack of robust evidence relating to the association between infection and telomere length.

While some infections have been studied in relation to telomere length, existing studies differ in the types and severities of infections studied, definitions used and use differing measures of telomere length, making pooling evidence across studies challenging.[9 11–15] Moreover, cross-sectional studies are the most abundant study type in this field; meaning there is a potential for reverse causality. Despite the heterogeneity, some evidence suggests that associations between some persistent viral infections such as cytomegalovirus and herpes simplex virus type-1 were associated with reduced telomere length or telomere attrition.[9 10 12 13 15] Current gaps in research include establishing whether infections as a whole are risk factors for reduced telomere length and whether pathogen type, severity and infection site are associated with telomere length. It is plausible that telomere attrition could act as a mechanism through which infections mediate effects on age-related diseases and thereby represent a target for intervention. However, the degree to which any associations are causal remains unclear A systematic review looking at the potential association between infection and telomere length is needed as no prior reviews have been conducted and they are crucial for identifying research gaps and informing the design of future studies.

## OBJECTIVES

This systematic review aims to comprehensively summarise all existing literature on the association between infections (by type, site, severity) and telomere length or attrition across a broad range of study designs (see eligibility criteria) in adult humans. We aim to establish whether there is an association between infection and telomere length to inform future studies.

### Research questions

1. Is there an association between infections and telomere length or attrition?
2. Is infection type, site, severity associated with telomere length or attrition?
3. Is preventing or treating infections associated with telomere length or attrition?

## Methods and analysis

The current protocol for our systematic review adheres to the guidelines provided by the Preferred Reporting Items for Systematic Reviews and Meta-analyses Protocols (PRISMA-P) statement and has been registered in PROS-PERO.[16 17] Any modifications to the protocol will be documented and updated on PROSPERO.

We intend to follow the PRISMA statement for reporting the systematic review and if applicable employ the Meta-analysis of Observational Studies in Epidemiology statement for the reporting of any potential meta-analysis.[18 19]

### Search strategy

We will perform a comprehensive search strategy (search date 31 August 2023) encompassing both published studies and grey literature. Published studies will be sought from six electronic databases, namely MEDLINE (Ovid interface), EMBASE (Ovid interface), Web of Science, Scopus, Global Health and the Cochrane Library. For grey literature, the British Library of electronic theses databases (ETHOS) will be explored. Additionally, the reference lists of included papers will be manually searched to identify any additional relevant studies.

We constructed a preliminary MEDLINE search using three concepts namely (1) infections, (2) telomere length and (3) human study type, these search concepts were combined using the Boolean operator 'AND'. Our search involved combining keywords with database-specific subject headings and this search can be found in the online supplemental appendix. This search was created and translated across databases with support from a librarian at the London School of Hygiene & Tropical Medicine. No restrictions were placed on the geographical location, language or date of publication of the studies. We will aim to translate any potentially relevant non-English language studies. The full search strategy can be found in the online supplemental information.

### Selection process

We will use reference management software EndNote (V.X8.0.2) for storing our search results. We will conduct de-duplication using the automated feature and subsequently inspect the results to identify and eliminate any duplicate entries manually.

For the selection of studies, two researchers will independently assess all titles and abstracts to determine their agreement with the eligibility criteria described below. Reviewers will discuss and agree on which articles should go to full-text review. We will then obtain the full texts and the process will be repeated. In the event of discrepancies between the reviewers, we will discuss these and if necessary a third reviewer will be consulted. All reasons for excluding studies will be documented at the full-text review stage and our study selection process will be illustrated using the PRISMA flow diagram.[18] If multiple papers stem from the same study population then we will include the paper that encompasses the largest sample

size and provides the most comprehensive exposure and outcome details.

## Eligibility criteria

Studies will be eligible for inclusion in the present study if they meet the criteria below:

### Study characteristics

To capture all potentially relevant designs, we will include cross-sectional studies, case—control studies, cohort studies, randomised control trials (of vaccination or infection treatment) and Mendelian randomisation studies.

We will include studies of any setting and time frame.

### Population

We will include studies with adults aged ≥18 years from any geographical area and any healthcare/study environment. Animal studies will be excluded.

### Exposure

The exposure group will be individuals exposed to infection (ie, any pathogen, site, severity, type, eg, acute or chronic). Infection diagnosis could be defined through electronic healthcare records (eg, using International Classification of Diseases, 10th revision (ICD-10) or Read coded diagnoses), self-report, antibody measures or other laboratory markers of infection. For Mendelian randomisation studies, exposure will be individuals who carry the genetic variants associated with infection and for randomised controlled trials the exposure would be people receiving a vaccine or treatment for infection.

### Comparators

The comparator group will vary by study type. For cross-sectional and cohort studies the comparator group will be individuals unexposed to infection. For case–control studies the comparator group is individuals with normal telomere length. For Mendelian randomisation studies, comparators will be individuals who do not carry the genetic variants associated with infection. Finally, for randomised controlled trials the comparator would be people not receiving vaccine or treatment for infection.

### Outcome

The outcome will be (1) telomere length and (2) telomere attrition for longitudinal studies. We will be inclusive in how the outcome is measured for example; measured by the rate of change, continuous measures or binary measures.

We will include studies with any valid method of ascertainment measurement of telomere length. These include PCR (Polymerase Chain Reaction) methods, TRF (Terminal Restriction Fragment) analysis, a variety of FISH (Fluorescence In Situ Hybridisation) methods, STELA (Single TElomere Length Analysis) and TeSLA (Telomere Shortest Length Assay).[20] We will not be limited by the cell type in which telomere length is measured.

### Data collection process

Two independent researchers will extract information from the selected papers using a piloted data extraction form. The first reviewer will conduct the data extraction in full whereas the second reviewer will extract data on a 10% random sample of the selected studies. In cases where essential data are missing, we will contact authors to request the necessary information.

### Data items

To create our data extraction form, we will adopt the Population, Exposure, Comparator, Outcomes and Study Characteristics framework.[21] Our data extraction will encompass the following elements:

1. Population: This section will include information about the population under study, such as age (mean, median or range), gender distribution and the criteria used for inclusion and exclusion, for example, health conditions, location of residence.
2. Exposure: We will extract details regarding the definition of the exposure, the type of infection involved, whether it relates to hospitalised infection, its acute or chronic nature and the number of individuals exposed.
3. Comparators: Information related to comparators will encompass their identification, definition and the count of comparators used in the study.
4. Outcomes: We will collect data on the type of measurement used for telomere length (eg, binary or continuous) and the number of participants who experienced the specified outcome.
5. Study Characteristics: This section will provide essential details about the study, including the authors' names, the study's title, publication year, study design, healthcare setting, country where the study was conducted, sample size and the duration of follow-up.

Furthermore, we will document any collected covariates and effect modifiers and ensure that both unadjusted and adjusted effect estimates and accompanying 95% CIs are included in our data extraction process. We will also include the results of subgroup analyses. for example, by age and sex.

### Assessing study bias

Two researchers will independently assess bias following the Cochrane collaboration approach[22] using the Risk Of Bias In Non-randomized Studies - of Exposure (ROBINS-E tool)[23] for observational studies and the revised tool for assessing risk of bias in randomised trials (ROB2 tool)[24] for randomised controlled trials. Both tools will be pilot tested. The ROBINS-E tool will involve evaluating the risk of bias in the following domains: confounding, measurement of the exposure, selection of participants into the study (or into the analysis), post-exposure interventions, missing data, measurement of the outcome, selection of the reported result. The ROB2 tool will involve evaluating bias related to the following domains: the

randomisation process, deviations from intended interventions, missing outcome data, measurement of the outcome, selection of the reported result.

## Data synthesis

We will categorise studies based on exposure (stratifying by factors such as infection/pathogen type and site, severity, acute or chronic status), outcome (such as length or attrition, cell type), study type and summarise data in predefined tables. Our primary analyses will focus on the main exposures of any infection, any vaccination and any antimicrobial treatment. We will then conduct secondary analyses of infection type and severity.

A meta-analysis will be considered feasible if there are at least five homogeneous studies in terms of design, exposure (infection/pathogen type, severity), outcome (telomere length measurement technique) as well as the time between exposure and outcome measurement. Pooled effect measures (ORs, risk ratios or HRs and corresponding 95% CIs) of the studies will be computed and study results displayed in Forest plots.

Statistical heterogeneity will be assessed using forest plots, $\chi^2$ test and $I^2$ statistic and a random effects meta-analysis will be conducted.[25 26] Publication bias and small study effects will be assessed with funnel plots if there are ≥10 eligible studies. If a meta-analysis is unfeasible then a narrative synthesis will be provided with results grouped by exposure.

## Certainty assessment

We will use the Grading of Recommendations Assessment, Development and Evaluation tool to evaluate evidence quality for each outcome.[27] Domains considered include risk of bias (determined as described in the 'Assessing study bias' section above), inconsistency, indirectness, imprecision and publication bias. The evidence will be categorised as high, moderate, low or very low.

## Ethics and dissemination

The present study does not require ethical approval. Results will be submitted for publication in a peer-reviewed journal and may be presented at relevant conferences. The review will highlight research gaps and future directions in this field.

## Patient and public involvement

Patients and the public were not involved in any way.

**Contributors** LT wrote the present paper and constructed the search strategy with the help of a London School of Hygiene & Tropical Medicine librarian. The writing and search strategy was reviewed by CW-G, RM, LH and JWB. LT will be conducting the search, data extraction as well as risk of bias and certainty assessments alongside another independent reviewer namely KAB.

**Funding** This work was supported by the Wellcome Trust. Grant number: Wellcome Career Development Award 225868/Z/22/Z to CW-G.

**Competing interests** None declared.

**Patient and public involvement** Patients and/or the public were not involved in the design, or conduct, or reporting, or dissemination plans of this research.

**Patient consent for publication** Not applicable.

**Provenance and peer review** Not commissioned; externally peer reviewed.

**ORCID iDs**
Louis Tunnicliffe http://orcid.org/0000-0001-8537-2855
Rutendo Muzambi http://orcid.org/0000-0003-0732-131X
Charlotte Warren-Gash http://orcid.org/0000-0003-4524-3180

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
