## [Reviewer comments · BMJ Open]

ARTICLE DETAILS

TITLE (PROVISIONAL)	Infection and telomere length: a systematic review protocol
AUTHORS	Tunncliffe, Louis; Muzambi, Rutendo; Bartlett, Jonathan; Howe, Laura; Abdul Basit, Khalid; Warren-Gash, Charlotte

VERSION 1 – REVIEW

REVIEWER	Kazeminasab, Somayeh Tabriz University of Medical Sciences
REVIEW RETURNED	26-Dec-2023

GENERAL COMMENTS	Understanding the causes of inter-individual variations infectious diseases could provide clues to the development of the personalized therapeutic intervention. The current knowledge highlights the significant association between the shorter telomere length and the higher risk of developing infectious diseases. This review indicates the detailed role of telomere length in infectious diseases. *The current coronavirus disease 2019 (COVID-19) pandemic has faced the world with unprecedented challenges. Please make sure that you include the most recent literature available in COVID-19.
--

REVIEWER	Topiwala, Anya University of Oxford, Nuffield Department of Population Health
REVIEW RETURNED	07-Feb-2024

GENERAL COMMENTS	This is a very clear and comprehensive protocol. The planned study is broad in scope which is appropriate for this understudied question, and I expect the results will be manageable. I had just a couple of queries: 1. Throughout the authors talk about "telomere length" but I was not clear if they were limiting their review to specific tissues or cell types e.g. leucocyte TL.2. Covariates - there are many important covariates e.g. alcohol, smoking, ethnicity that have been implicated in TL that I think it would be important to examine, particularly when comparing associations from different studies. Furthermore, blood constituents e.g. neutrophil/leucocyte counts could be important when investigating links with infection i.e. are any associations the result of TL shortening from infection or could they reflect changes in blood composition as a result of infection?
---

VERSION 1 – AUTHOR RESPONSE

Reviewer: 1

Dr. Somayeh Kazeminasab, Tabriz University of Medical Sciences

Comments to the Author:

Understanding the causes of inter-individual variations infectious diseases could provide clues to the development of the personalized therapeutic intervention. The current knowledge highlights the significant association between the shorter telomere length and the higher risk of developing infectious diseases. This review indicates the detailed role of telomere length in infectious diseases.

*The current coronavirus disease 2019 (COVID-19) pandemic has faced the world with unprecedented challenges. Please make sure that you include the most recent literature available in COVID-19.

Response:

- We thank the reviewer for their positive comments on our protocol. We agree that the potential association between COVID-19 and telomere length is important to capture in our systematic review. Our search was conducted as of August 31, 2023 with broad terms to capture all infections including SARS-COV-2 so we are confident that these studies will be included. We have now specified our search date in the 'Search strategy' section, the revised text is below in italics (see attached response document):

'We will perform a comprehensive search strategy (search date August 31, 2023) encompassing both published studies and grey literature. Published studies will be sought from six electronic databases, namely MEDLINE (Ovid interface), EMBASE (Ovid interface), Web of Science, Scopus, Global Health, and the Cochrane Library. For grey literature the British Library of electronic theses databases (ETHOS) will be explored. Additionally, the reference lists of included papers will be manually searched to identify any additional relevant studies.'

- Reviewer: 2

Dr. Anya Topiwala, University of Oxford

Comments to the Author:

This is a very clear and comprehensive protocol. The planned study is broad in scope which is appropriate for this understudied question, and I expect the results will be manageable. I had just a couple of queries:

1. Throughout the authors talk about "telomere length" but I was not clear if they were limiting their review to specific tissues or cell types e.g. leucocyte TL.

Response:

- We have now updated our protocol (Eligibility criteria section) to clarify that we aim to include literature regarding telomere length in any cell type. When presenting results, we will group similar studies together by factors including study design, exposure and outcome type. Outcome will be sub-categorised further into cell type. The revised text is below in italics (see attached response document):

'We will include studies with any valid method of ascertainment measurement of telomere length. These include PCR (Polymerase Chain Reaction) methods, TRF (Terminal Restriction Fragment) analysis, a variety of FISH (Fluorescence In Situ Hybridization) methods, STELA (Single TELomere

Length Analysis), and TeSLA (Telomere Shortest Length Assay) (21). We will not limit by the cell type in which telomere length is measured.'

2. Covariates - there are many important covariates e.g. alcohol, smoking, ethnicity that have been implicated in TL that I think it would be important to examine, particularly when comparing associations from different studies. Furthermore, blood constituents e.g. neutrophil/leucocyte counts could be important when investigating links with infection i.e. are any associations the result of TL shortening from infection or could they reflect changes in blood composition as a result of infection?

Response:

- We agree that capturing data on covariates such as alcohol, smoking and ethnicity is important to assess the likelihood of residual confounding. We will extract data on all covariates assessed in the included studies, which is described in the 'Data items' section of our protocol. In addition, we will ensure that adjustment for covariates is assessed during our risk of bias assessment and will consider residual confounding as a potential limitation when interpreting findings from included studies. Unfortunately, it is unlikely that blood composition will be captured in the included studies. However, we will carefully interpret our findings taking into account factors such as the cell type in which telomeres were measured, whether an infection is acute or chronic as well as infection severity and pathogen, which will help us to clarify the nature and potential mechanisms of any association seen.

VERSION 2 – REVIEW

REVIEWER	Topiwala, Anya University of Oxford, Nuffield Department of Population Health
REVIEW RETURNED	25-Mar-2024
GENERAL COMMENTS	The authors have addressed all my previous comments in full. I have no further concerns.

VERSION 2 – AUTHOR RESPONSE